# GNL3 and PA2G4 as Prognostic Biomarkers in Prostate Cancer

**DOI:** 10.3390/cancers15102723

**Published:** 2023-05-11

**Authors:** Shashank Kumar, Mohd Shuaib, Abdullah F. AlAsmari, Faleh Alqahtani, Sanjay Gupta

**Affiliations:** 1Molecular Signaling & Drug Discovery Laboratory, Department of Biochemistry, Central University of Punjab, Guddha, Bathinda 151401, Punjab, India; mohd.shuaib9519@gmail.com; 2Department of Pharmacology and Toxicology, College of Pharmacy, King Saud University, Riyadh 11451, Saudi Arabia; afalasmari@ksu.edu.sa (A.F.A.); afaleh@ksu.edu.sa (F.A.); 3Department of Urology, School of Medicine, Case Western Reserve University, Cleveland, OH 44106, USA

**Keywords:** prostate cancer, biomarkers, overall survival, TCGA database, prognostics

## Abstract

**Simple Summary:**

Guanine nucleotide-binding protein-like 3 (GNL3) and proliferation-associated protein 2G4 (PA2G4) are molecules involved during metaphase-to-anaphase transition and growth regulation. GNL3 and PA2G4 have been found to be overexpressed in several human cancers, including prostate cancer. Clinical data suggest that GNL3 and PA2G4 could be developed as prognostic biomarkers of clinical significance in prostate cancer. This review article mainly highlights the function of GNL3 and PA2G4 and focuses on the opportunities for their development as prognostic biomarkers in prostate cancer.

**Abstract:**

Prostate cancer is a multifocal and heterogeneous disease common in males and remains the fifth leading cause of cancer-related deaths worldwide. The prognosis of prostate cancer is variable and based on the degree of cancer and its stage at the time of diagnosis. Existing biomarkers for the prognosis of prostate cancer are unreliable and lacks specificity and sensitivity in guiding clinical decision. There is need to search for novel biomarkers having prognostic and predictive capabilities in guiding clinical outcomes. Using a bioinformatics approach, we predicted GNL3 and PA2G4 as biomarkers of prognostic significance in prostate cancer. A progressive increase in the expression of GNL3 and PA2G4 was observed during cancer progression having significant association with poor survival in prostate cancer patients. The Receiver Operating Characteristics of both genes showed improved area under the curve against sensitivity versus specificity in the pooled samples from three different GSE datasets. Overall, our analysis predicted GNL3 and PA2G4 as prognostic biomarkers of clinical significance in prostate cancer.

## 1. Introduction

Prostate cancer is a prevalent disease in males worldwide with an existing rate of 29% for all diagnosed cancers and the fifth leading cause of cancer-related deaths [1]. According to Globocan 2020, approximately 1.14 million prostate cancer cases were diagnosed and 0.375 million deaths occurred in 2020 [2]. Prostate cancer is a heterogeneous disease, ranging from remarkably low-aggressive, organ-confined to high-aggressive, non-organ confined lethal phenotypes. The therapeutic decision and survival outcome of prostate cancer is dependent on appropriate patient stratification to different risk groups; therefore, it is very important to differentiate between indolent and aggressive diseases. Clinical diagnosis and prognosis of prostate cancer is currently based on digital rectal examination (DRE), serum levels of prostate-specific antigen (PSA), and pathologic Gleason score. The PSA test as a screening tool for prostate cancer was first approved by the Food and Drug Administration in 1986, and is still controversial because of high false-positive rates and the risks associated with biopsies and over-treatment [3]. As such, PSA is a non-specific biomarker for prostate cancer, and its expression has also been reported in other organs such as the adrenals, small intestine, kidney, and salivary tissue [4]. There is a strong debate regarding PSA as a diagnostic and prognostic marker since it is unable to differentiate between indolent and aggressive forms of prostate cancer. This is evidenced by the fact that many men harbor aggressive prostate cancer while displaying low levels of serum PSA. Moreover, the Gleason grading system used with prostate biopsy specimens to evaluate the clinical progression of men with prostate cancer needs further refinement for more accurate grade stratification. Together with these parameters, it is especially important to focus on other types of molecular markers that can support clinical outcomes and disease prognostication [5,6].

In the past decade more advancements in genomic, epigenomic, proteomic, and bioinformatics-based techniques have been implicated, which are helpful in identifying potential diagnostics and prognostic biomarkers for prostate cancer [7,8,9,10]. A biomarker is a measurable biological indicator that can provide information about the incidence or progression of a disease or the effects of an undertaken treatment [11]. A clinically relevant biomarker should be safe and obtained through a non-invasive protocol, have high sensitivity and specificity and high positive and negative predictive values, and make possible clinical decisions [11]. In recent years, a number of clinical and biological prognostic biomarkers of prostate cancer have been reported, such as androgen receptor variant V7 [12], PTEN [13], or c-MYC gene inactivation [14], although, none of them have been approved as a prognostic biomarker for use in the clinical settings [15]. Other biomarkers such as TMPRSS2-ERG gene fusion [16], non-coding RNA (PCA3) [17,18], and kallikrein included in basic PHI (prostate health index) or 4K tests have shown to increase sensitivity and specificity of serum PSA in prostate cancer patients. Pathogenic variants in genes such as BRCA1 and BRCA2 mismatch repair genes [19] and HOXB13 [20] confer modest to moderate lifetime risk of prostate cancer. Notably, BRCA2 has emerged for its clinical relevance in the treatment and screening of prostate cancer [19]. Genome-wide association studies (GWAS) have identified several single-nucleotide polymorphisms (SNPs) that are independently associated with incremental risk of prostate cancer [21]. Although more than 150 SNPs associated with prostate cancer have been identified, the clinical utility remains uncertain. Moreover, there are a few commercially available molecular prognostic biomarkers in clinical use such as OncotypeDX Genomic Prostate Score, Prolaris, ProMark, and Decipher, based on a gene set panel altered in prostate cancer (Table 1).

These molecular tests, recognized as companion diagnostics, help in guiding clinicians to establish an appropriate treatment strategy, and predict recurrence and progression risk [34]. More recent studies are directed towards the use of artificial intelligence and its algorithms for clinical applications in monitoring, detection, diagnosis, and treatment to generate new predictive models [35]. However, there is still a need for identification of precise molecular markers that can aid in early diagnosis and prognosis and establishment of response to various treatments in prostate cancer patients. The integration of clinically valuable potential prognostic biomarkers is very important to the treatment of prostate cancer as they suggest their potential utility in disease management. In the present review, we performed bioinformatic analysis to identify genes that are differentially expressed in prostate cancer compared to normal prostate utilizing TCGA dataset. We also performed Kaplan–Meier survival analysis, receiver operating characteristic analysis, protein–protein interaction, co-expression analysis, and elucidation of biological pathways on these datasets. Based on the published information and our bioinformatics approach, we propose GNL3 and PA2G4 as prognostic biomarkers in prostate cancer. 

## 2. Identification of Prognostic Biomarker(s) in Prostate Cancer

A total of 164 genes were identified as prostate cancer biomarkers of prognostic significance that were extracted using the ULCAN online tool (https://ualcan.path.uab.edu/analysis-surv.html, accessed on 3 March 2023). We checked the expression pattern of identified genes in prostate cancer specimens compared to normal prostate samples. A total of 30 biomarkers were selected based on their significant higher differential expression in prostate cancer compared to normal prostate tissue. Then, we sorted the top seven biomarker genes based on the grade-dependent (Gleason score 6–10) expression patterns in prostate cancer specimens compared to normal prostate tissue (TCGA database). Next, we performed a literature search to ensure the available reports on the prognostic relevance of these genes. Based on the information CHCHD8, GNL3, PA2G4, and RRP9 genes were selected to further explore the prognostic significance in prostate cancer. In our differential expression analysis, we found that CHCHD8, GNL3, PA2G4, and RRP9 have 27.0-, 38.6-, 12.8-, and 7.6-times higher expression in prostate cancer specimens in comparison to normal prostate tissue. The expression profile of CHCHD8, GNL3, PA2G4, and RRP9 genes in prostate cancer compared to normal tissue samples together with grade-specific expression based on Gleason score is shown in Figure 1A–D. 

The GNL3, CHCHD8, PA2G4, and RRP9 genes were selected to predict their ability in overall and disease-free survival in prostate cancer patients. For this, we performed survival analysis in prostate cancer specimens available in the TCGA database using the Kaplan–Meier survival method in the GEPIA2 online tool (http://gepia2.cancer-pku.cn/#index, accessed on 6 March 2023) with the hazard ratio (HR) of more than 1.0 criteria used to predict the potential of genes as prognostic biomarkers. We found that the higher expression of GNL3, CHCHD8, PA2G4, and RRP9 genes with HR more than 1.0 was closely associated with poor overall survival (Figure 2A–D) and disease-free survival in prostate cancer patients (Figure 3A–D).

Next, the Receiver Operating Characteristic (ROC) analysis was performed to evaluate the diagnostic potential of CHCHD8 GNL3, PA2G4, and RRP9 genes (Figure 4A–D). ROC curve analysis is a popular method to measure the diagnostic value of any biological marker in context of sensitivity, specificity, and area under curve (AUC). Several studies reported potential diagnostic biomarkers for various cancers using GSE database by performing a ROC analysis. Typically, the AUC value > 0.5 have significant difference between diseased and non-diseased stages. To perform the ROC analysis, we extracted the expression values of each gene from three different GSE datasets (GSE3325, GSE6919, and GSE55945) of prostate cancer specimens and normal prostate samples from NCBI-GEO-datasets. Then we performed the ROC analysis using the online ROC plotter statistics tool (http://www.rad.jhmi.edu/jeng/javarad/roc/JROCFITi.html, accessed on 9 March 2023) in that GSE datasets. We found that GNL3 and PA2G4 showed comparatively significant and better performance in prostate cancer specimens compared to normal samples (Figure 4B,C). The GNL3 and PA2G4 genes showed a significant AUC value compared to AUC of commercially available biomarkers (0.60–0.70) (Table 1).

## 3. Biological Pathways and Protein–Protein Interaction Analysis

Previous analysis identified GNL3 and PA2G4 as the lead genes after Kaplan–Meier survival and ROC analysis. Further, these two gene sets were chosen to study their involvement in biological pathways, protein–protein interaction (PPI), and co-expression pattern. We utilized FunRich, an offline software to determine the involvement of GNL3 and PA2G4 in different biological signaling pathways by applying a statistically significant *p* value less than 0.05. PPI and co-expression analysis was performed using PPI string database. We observed that GNL3 and PA2G4 were enriched in the regulation and co-regulation of androgen receptor (AR) activity, androgen receptor-mediated signaling, regulation of β-catenin signaling, and Wnt signaling pathways (Figure 5A–C).

The top ten biological pathways of GNL3 and PA2G4 with their gene count percentage and significant –log10 *p* value include syndecan-4 mediated signaling, glypican network, canonical and non-canonical Wnt signaling, β-catenin signaling, and androgen receptor (AR) signaling pathways (Figure 5A). Accumulating data suggests that AR-mediated signaling pathways play a crucial role in the development, progression, and resistance to antiandrogen therapy of prostate cancer. Accumulating evidence indicates a significant role of androgen receptor splice variants in mediating resistance of castration-resistant prostate cancer to current therapies and in predicting therapeutic responses [36,37]. Similarly, Wnt signaling components also play a significant role in prostate tumorigenesis and promote resistance against androgen deprivation therapy [38,39]. Both canonical and non-canonical Wnt signaling pathways regulate several developmental and biological processes including cell proliferation, self-renewal, and stem cell differentiation. In the non-canonical Wnt signaling pathway, Wnt5a, Wnt5b, and Wnt11 ligands bind to a panel of diverse receptors to activate Wnt signaling, including receptors of the Frizzled family and other mediators such as tyrosine-protein kinase transmembrane receptor (ROR1, ROR2 or RYK) [40]. Binding of these non-canonical Wnt ligands can activate multiple intracellular pathways including the planar cell polarity and calcium signaling pathways. The non-canonical Wnt signaling pathway plays a role in prostate cancer progression to an AR-indifferent or neuroendocrine phenotype where the Wnt secretion mediator, Wntless is recognized as a major driver of neuroendocrine-differentiated prostate cancer characterized by aggressive tumor growth [41]. These Wnt prostate tumors express minimal to low levels of AR and reduced PSA. The canonical Wnt signaling is dependent on β-catenin as an effector of Wnt proteins, and its high level induces tumorigenesis. In the absence of extracellular Wnt signals, cytoplasmic β-catenin is phosphorylated by glycogen synthase kinase 3 (GSK3) as part of a destruction complex including adenomatous polyposis coli (APC) and axin proteins. The phosphorylated β-catenin is then ubiquitinated and degraded. Wnt signaling inhibits this process leading to the accumulation of β-catenin in the nucleus by enabling the formation of transcriptionally active complexes [42]. Interaction of β-catenin and its crosstalk with AR has been well documented in prostate cancer. AR binds β-catenin directly to stimulate AR-mediated gene transcription that provides a growth advantage engaging downstream targets such as c-Myc and cyclin D1, even at the castration levels of androgens [43]. The PPI network analysis further revealed that GNL3 and PA2G4 have greater interaction with their co-expressed/co-occurred proteins (Figure 5B,C). In the co-expression analysis, we found that GNL3 protein strongly co-expressed with WDR12 and GNL2 proteins. Similarly, PA2G4 protein expression was closely associated with the expression of RPL4 protein (Figure 5C).

The coiled-coil-helix-coiled-coil-helix domain containing 8 (CHCHD8) is a putative COX assembly factor known as cytochrome c oxidase assembly factor 4 homolog (CoA4). COA4 is a newly identified CcO assembly factor which is a twin CX(9)C motif mitochondrial protein localized in the intermembrane space linked with the inner membrane of mitochondria. Its transport into intermembrane space depends on the MIA40 trans-site receptor machinery [44,45]. It is well-known that the mitochondria organelles are a leading source of cellular energy and reactive oxygen species (ROS) [45]. Preclinical and clinical studies have demonstrated that increased levels of ROS, especially free radicals, cause oxidative damage in DNA, proteins, and lipids which lead to the pathogenesis and the progression of prostate cancer. Cytochrome c oxidase (CcO) is an enzyme in the mitochondrial respiratory chain that powers cellular energy production as ATP. CcO enzyme is a tightly regulated protein that is involved in mitochondrial mediated oxidative metabolism, phosphorylation, and ATP formation. In general, the CoA4 protein plays a role in CcO assembly for the mitochondrial respiratory chain. Therefore, deletion and a mutation in the CoA4 protein impair the assembly of CcO, which induces the amplification of hydrogen peroxide production and prevents cell proliferation in a normal condition [46]. Moreover, it has also been shown that CoA4-lacking cells suppress CcO activity [25]. The suppressed activity of CcO has been implicated in the metabolic shift towards glycolysis, and defects in the assembly of the CcO complex lead to the induction of Ca^2+^/Calcineurin-mediated retrograde signaling. Ca^2+^/Calcineurin-mediated retrograde signaling can activate PI3-kinase, IGF1R, and AKT. It is well documented that these proteins are involved in the oncogenic transformation and cancer progression [47]. However, the role of CoA4 is limited in cancer. Krobthong et al. (2022) reported that cancer-promoting proteins, including CoA4, were downregulated in the natural peptides treated A549 lung cancer cells compared to the non-treated cells. The authors describe that natural peptides have anti-proliferative and anti-metastatic activities by suppressing cancer-promoting proteins and further reported that the anti-oxidative activity of natural peptides may be attributed to higher expression of ROS-reducing proteins [48]. This study indicates that CoA4 may be involved in ROS production.

The Guanine nucleotide-binding protein-like 3 (GNL3) is alternatively known as nucleostemin. In general, GNL3 stabilizes the telomeric repeat binding factor 1 (TRF1) protein during the processes of mitosis and stimulates the metaphase-to-anaphase transition. GNL3-mediated stabilization of TRF1 controls the telomere and cell cycle progression [49]. Telomeres are the protective structures of chromosome ends that are gradually shortened by each cell division, eventually leading to cellular senescence. Malignant cells maintain the telomere length for unlimited growth by telomerase reactivation or a recombination-based mechanism. Therefore, telomere length has emerged as an emerging therapeutic target in majority of human cancers. Elevated expression of GNL3 in various cancers has been detected. Several studies reported that elevated expression of GNL3 protein promotes cell proliferation, invasion, migration, and epithelial-to-mesenchymal transition in several cancers, including prostate cancer [50,51,52]. It has been reported that loss of GNL3 expression inhibits cell proliferation, migration, invasion and induces apoptosis in various types of cancer cells [53,54]. Recently, Zhang et al. (2022) found the overexpression of GNL3 protein promotes malignant behavior of liver cancer cells. The authors further demonstrated that knockdown of GNL3 inhibits proliferation, migration, and invasion of liver carcinoma cells. These results highlight that aberrant expression of GNL3 is associated with poor overall survival of hepatocellular carcinoma patients [44]. Another in vitro study found that overexpression of GNL3 protein accelerates epithelial-to-mesenchymal transition and decreased expression of GNL3, thereby lowering growth, migration, and invasion of osteosarcoma cells [54]. Similarly, Dai et al. (2021) showed that upregulated expression of GNL3 promotes non-Hodgkin lymphoma progression by stimulating the oncogenic Wnt/β-catenin signaling [55]. Another study found that elevated expression of GNL3 with STAT3 activation is an early process in the progression of low-grade dysplasia squamous cell carcinoma [56]. Sami et al. (2019) demonstrated that nucleostemin (GNL3) protein is a predicted biomarker in the most aggressive phenotype of breast cancer [57]. Lin et al. (2019) revealed that higher nucleostemin expression is associated with poor progression-free survival in triple-negative/basal-like breast cancers [58]. Few other studies found upregulation of GNL3 in prostate cancer. Liu et al. (2009) reported that nucleostemin acts as a critical G1/S barrier regulator, and its higher expression promotes prostate cancer progression [59]. Another study reported that higher expression of nucleostemin at the mRNA and protein level in prostate cancer cells and tissues and that decrease in its expression inhibits PC-3 cell proliferation [60]. Furthermore, a study reported that higher nucleostemin expression enhances prostate cancer tissues’ malignant behavior [35]. Various in vitro studies reported that decreased expression of GNL3 noticeably reduces proliferation in prostate cancer cells [52,60,61,62].

The proliferation-associated protein 2G4 (PA2G4) is an ErbB3-binding protein 1 (EBP1). EBP1 (PA2G4) is ubiquitously expressed in the nucleus and cytoplasm of malignant and non-malignant cells [63,64]. Cytoplasmic Ebp1 protein binds with the cytoplasmic domain of ErbB3 protein and leads to activation of ErbB3 that has been linked with several human cancers, and it has also been reported that ErbB3 signaling plays a crucial role in prostate cancer [63,65]. However, elevated expression of the PA2G4 gene has also been associated with the prognosis of various cancers including hepatocellular carcinoma, nasopharyngeal carcinoma, neuroblastoma, breast, and pancreatic cancers [63,66]. Several studies reported that higher expression of PA2G4 protein promotes cell proliferation, invasion, and migration and inhibits apoptosis in various cancer cells. Recently, Sun et al. (2022) found differential expression of PA2G4 in hepatocellular carcinoma compared to normal liver tissue and reported that the overexpression of PA2G4 accelerates epithelial-to-mesenchymal transition in hepatocellular carcinoma via stabilizing the mRNA of FYN oncogene and its overexpression is associated with poor prognosis of hepatocellular carcinoma patients [67]. Another study reported that higher expression of PA2G4 protein is responsible for poor outcomes in nasopharyngeal carcinoma. Further studies demonstrated that elevated expression of PA2G4 is closely associated with poor survival of nasopharyngeal carcinoma patients and acts as an independent prognostic indicator in the survival of these patients [68]. Liu et al. (2015) reported that high levels of Ebp1 are positively associated with the TNM stages of cervical cancer and involved in lymphatic metastasis [69]. Hou et al. (2021) reported that silencing of the PA2G4 gene repressed cell viability and induced apoptosis by activating the expression of caspase-3 and caspase-9 in glioblastoma cells [70]. Few studies reported overexpression of Ebp1 (PA2G4) protein in prostate cancer. It is known that the ErbB3 protein receptor is a crucial regulator in prostate cancer progression, and it has been demonstrated that binding of Ebp1 with ErbB3 protein is involved in their activation. Loss of expression of Ebp1 protein leads to deactivation of ErbB2/3 signaling, which significantly suppresses the growth of castration-resistant prostate cancer xenografts [71,72,73]. Ectopic expression of Ebp1 regulates cellular proliferation and enhances differentiation in prostate cancer cells [74]. Gannon et al. (2008) demonstrated that high expression of Ebp1 is directly associated with prostate cancer and progressively involved in the transition from the hormone-sensitive to the hormone-refractory stage. Furthermore, in vitro and in vivo study revealed that overexpression of Ebp1 causes therapeutic resistance, and its downregulation increases the sensitivity towards lapatinib in prostate cancer [75].

The Ribosomal RNA processing9 (RRP9) is a U3 small nucleolar RNA binding protein. The Rrp9 protein consists of a WD repeat domain and an N-terminal region, and these WD repeat domains are commonly involved in the interactions between the proteins [76]. Besides WD repeat domain, RRP9 also has another domain that is β-propeller that is composed of seven WD subdomains and mediate the interaction of RRP9 within the small subunit (SSU)-processome. The SSU-processome is a large ribonucleoprotein that is required for the assemblage of the SSU of the ribosomes and for the activation of the 18S rRNA, and the SSU processome is basically responsible for cell survival. RRP9 protein is involved in the functioning of U3 small nucleolar RNA, and U3 and RRP9 are responsible for 18S rRNA production by the SSU processome complex required for early pre-rRNA cleavages at the sites of A0, A1, and A2 [77]. Mutated RRP9 suppresses cell growth, while WT RRP9 restored yeast cell growth [77]. A recent study reported that a RRP9 neddylation deficit prevents pre-rRNA processing and leads to the downregulation of ribosomal biogenesis. Some studies suggested that overexpression of RRP9 enhances tumor cell proliferation, colony formation, and cell migration [78]. Moreover, ribosomal RNA biogenesis has been associated with various human malignancies [79]. Zhang et al. (2022) explored the role of U3 small nucleolar RNA binding protein RRP9 in pancreatic cancer. They reported that the RRP9 protein activates the AKT signaling pathway by binding with the DNA binding area of IGF2BP1 in pancreatic cancer cells, accelerating cancer progression, inhibiting apoptosis, and causes resistance against gemcitabine via decrease in DNA damage. It is reported that the expression of RRP9 is inversely correlated with the prognosis of pancreatic cancer patients [79]. Du et al. (2022) explored the oncogenic role of RRP9 in colon cancer. They found that elevated expression of RRP9 is involved in colorectal cancer tumorigenesis and its progression, and that the loss of RRP9 inhibits cell proliferation and migration, promotes tumor cell senescence, and halts tumor growth in nude mice xenografts [78].

There are few limitations in the study. Firstly, our analysis is mainly associated with the acquired information from various datasets and resources, and the results were not validated at a further biological experimental level. Next, our research was limited to the selection of candidate biomarker associated with the pathogenesis and prognosis of prostate cancer, which may lead to the negligence of some information. Finally, our study focused on the gene subsets which were identified having significant expression level change between cancer and non-cancer specimens in multiple datasets. Further studies are needed to compare GNL3 and PA2G4 expression with other established prostate cancer markers in terms of accuracy and probability.

## 4. Conclusions

In summary, GNL3 and PA2G4 play crucial biological roles in cancer, supporting important cellular processes such as cell migration, proliferation, apoptosis, and tumor growth. Functional studies on GNL3 and PA2G4 demonstrate a significant biological connection between their expressions in regulating various biological functions. In prostate cancer, aberrant expression of GNL3 and PA2G4 correlate with tumorigenesis and metastasis. Based on our analysis, GNL3 and PA2G4 showed better performance as prognostic biomarkers and may be used separately or in combination in prostate cancer. However, additional research is still required.

## Figures and Tables

**Figure 1 cancers-15-02723-f001:**
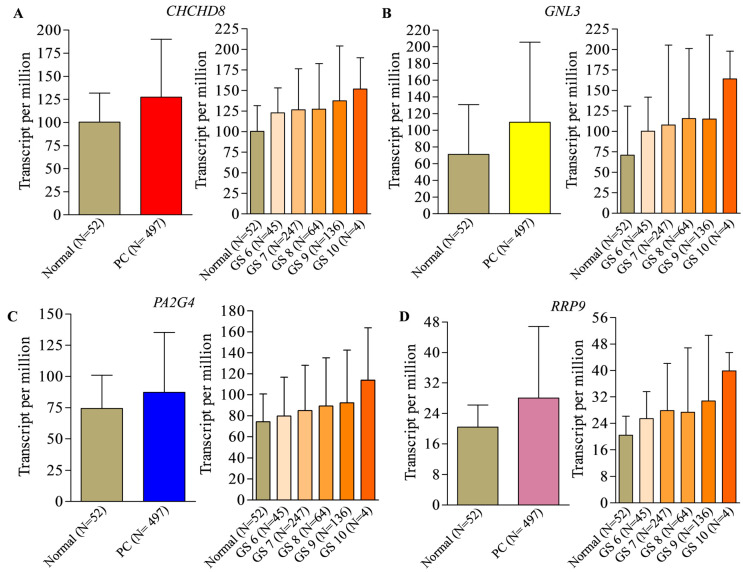
Expression of genes with prognostic significance in prostate cancer patients derived from TCGA dataset. Expression of (**A**) CHCHD8 (**B**) GNL3 (**C**) PA2G4 and (**D**) RRP9 in prostate cancer with their Gleason scores and normal prostate tissue samples. CHCHD8, Coiled-coil-helix-coiled-coil-helix domain containing 8, GNL3, Guanine nucleotide-binding protein-like 3, PA2G4, Proliferation-associated protein 2G4, and RRP9, Ribosomal RNA processing 9.

**Figure 2 cancers-15-02723-f002:**
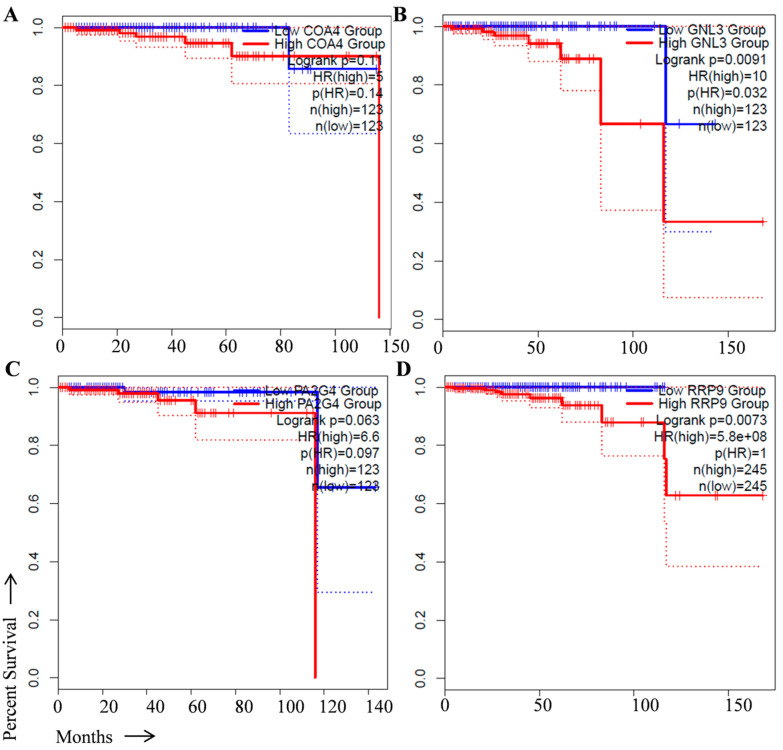
Overall survival analysis of signature genes in prostate cancer patients from TCGA dataset. Effect of (**A**) CHCHD8 (known as COA4), (**B**) GNL3, (**C**) PA2G4, and (**D**) RRP9 expression on overall survival in prostate cancer patients. The log rank values for CHCHD8 *p* = 0.11; GNL3 *p* = 0.0091; PA2G4 *p* = 0.063; and RRP9 *p* = 0.0073, respectively. Blue line, low gene expression; Red line, high gene expression.

**Figure 3 cancers-15-02723-f003:**
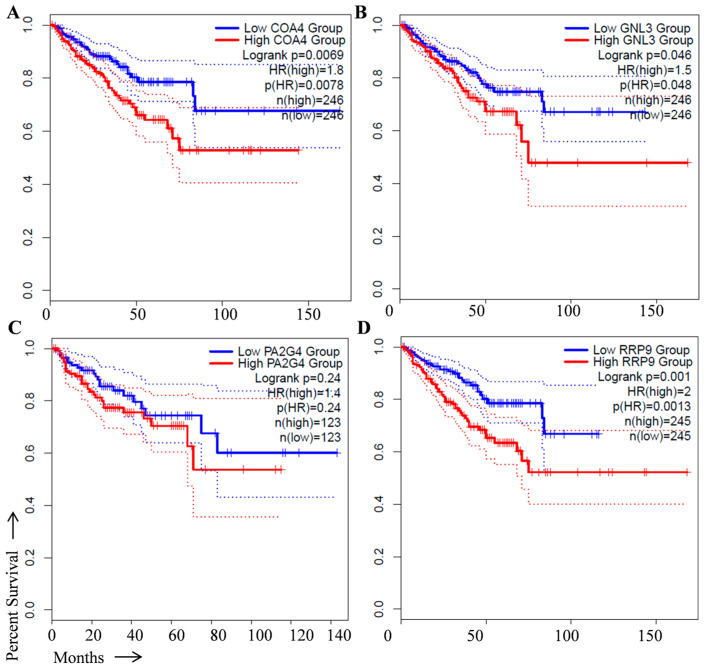
Disease free survival analysis of signature genes in prostate cancer patients from TCGA dataset. Effect of (**A**) CHCHD8 (known as COA4), (**B**) GNL3, (**C**) PA2G4, and (**D**) RRP9 expression on disease free survival in prostate cancer patients. The log rank values for CHCHD8 *p* = 0.0069; GNL3 *p* = 0.046; PA2G4 *p* = 0.24; and RRP9 *p* = 0.001, respectively. Blue line, low gene expression; Red line, high gene expression.

**Figure 4 cancers-15-02723-f004:**
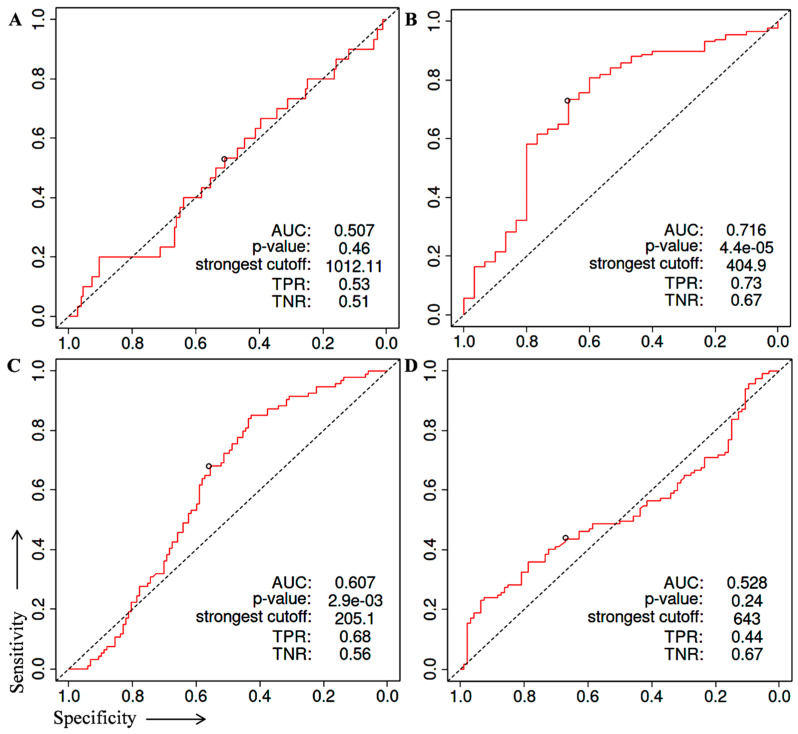
Receiver operating characteristic curve (ROC) analysis in normal tissue and prostate cancer specimens from GSE3325, GSE6919 and GSE55945 datasets. (**A**) ROC curve of CHCHD8 (AUC 0.507), (**B**) ROC curve of GNL3 (AUC 0.716), (**C**) ROC curve of PA2G4 (AUC 0.607), (**D**) ROC curve of RRP9 (AUC 0.528), respectively.

**Figure 5 cancers-15-02723-f005:**
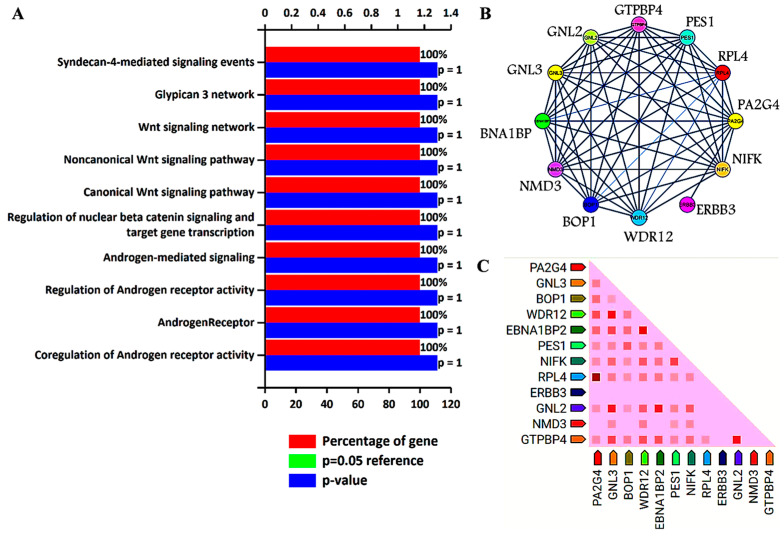
Biological signaling pathway enrichment and protein–protein interaction analysis of GNL3 and PA2G4 genes. (**A**) Top ten biological signaling pathways of GNL3 and PA2G4. Green bar is invisible as it overlaps with the blue bar. (**B**) Protein–protein interaction of GNL3, and PA2G4. GNL3 and PA2G4 are displayed in yellow circle. (**C**) Observed co-expression of GNL3 and PA2G4 in humans.

**Table 1 cancers-15-02723-t001:** Commercially available kits for prostate cancer diagnosis and prognosis.

Panel	Specimen	AUC for Prostate Cancer Detection	Limitations	Refs.
PCA3	Urine	0.65	Recommended only for the precise population of prostate cancer patients who have a first negative biopsy report. PCA3 score rises with age, independent of PC occurrence.	[17,18]
PHI	Serum	0.70	Used to detect the probability of finding any prostate cancer on repeat biopsy, irrespective of the GS. It does not have de facto common use because of pre-analytical stability of [2] proPSA and high cost.	[22,23]
4Kscore^®^	Serum	0.71	4Kscore^®^ test has been restricted to those prostate cancer patients who have not had DRE in the previous 96 h.	[24]
ConfirmMDx^®^	Prostate biopsy tissues	0.74	Recommended for those prostate cancer patients who had negative prostate biopsy. There is no recommendation for the routine clinical application.	[25]
ExoDX Prostate IntelliScore	Urine	0.70	Lack of evidence in respect of clinical utility	[26]
Prostate Core Mitomic Test	Serum	-	False negative results in malice of high sensitivity.	[27,28]
MiPS Mi(chigan) Prostate Scor	Urine	0.69	Lack of evidence in respect of clinical utility	[26]
TMPRSS2-ERG gene fusion	Urine	-	Not widely used due to long processing time, very high cost, and the necessity of dedicated equipment.	[28]
SelectMDX	Urine	0.71–0.81	Diagnostic as well as prognostic correctness in ethnically diverse study population is unidentified till date. SelectMDx shows declined sensitivity, specificity, and NPV.	[29,30]
Prolaris	Tissue	0.78	There is no evidence to describe the effect of the prolaris cell cycle progression test on patient-important clinical outcome results.	[30,31]
OncotypeDx	Tissue	0.73	This test is not intended to take ethnic discrimination into account.	[32,33]
ProMark	Tissue	0.72	Test is imperfect and skips the high-risk nearby zone of prostate tumor.	[26,33]

AUC: Area under curve; PC: Prostate cancer; PHI: Prostate health index; Digital rectal exam (DRE); PCA3: Prostate cancer antigen 3; NPV: Negative predictive value.

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
