# Peer review of "GNL3 and PA2G4 as Prognostic Biomarkers in Prostate Cancer"

_cancers, 2023, doi:10.3390/cancers15102723_

Round 1

Reviewer 1 Report

Overall a well-written technical content of the manuscript.

Providing a document with marked notes of critique and possible edits. See the attached sticky notes. Several statements need to be referenced.

Figure 2 panels are very crowded to read- can these be decluttered by moving the log rank information to the legend for each plot?

The technical English was good, however the grammar and punctuation needed substantial editing. Most of the edits are grammatical in nature; although some sentences are not clear as written and clarity was requested.

Author Response

We thank the reviewer for critical evaluation of the manuscript. We have revised the manuscript and have added additional references as per suggestion.

Figure 2, the point is well taken. We have revised the figure which is now divided in two figures for better clarity and visibility. The log rank for each gene is provided in the figure legend.

Reviewer 2 Report

The authors should be congratulated for the interesting topic discussed. 

Despite the countless successes in the clinical field and in the treatment approach of PCa, it is still challenging due to the not yet universally understood genomic characteristics and the lack of prognostic biomarkers of clinical significance.

I believe that the study has sufficient merit to be considered for publication, although major revisions are required. 

1.     A lecture of this paper, https://doi.org/10.3390/diagnostics11050908 is recommended. In the introduction, authors should even discuss the genetic mechanism underlying development of PCa, especially regarding the role of BRCA.

2.    Please, the authors should discuss the limitations of this study and how these findings could enhance clinical practice. 

3.    The quality of panels of Figure 2 could be implemented.

There is a mistake in citation no. 9 and 10. These two overlap. Please, separate them.

Minor editing of English language required,

Author Response

A lecture of this paper, https://doi.org/10.3390/diagnostics11050908 is recommended. In the introduction, authors should even discuss the genetic mechanism underlying development of PCa, especially regarding the role of BRCA.

The point is well taken, and we have incorporated the information on BRCA1/2 in the manuscript with citation.

Please, the authors should discuss the limitations of this study and how these findings could enhance clinical practice.

A paragraph on the limitation of the study has been included in the revised manuscript as suggested by the reviewer.

The quality of panels of Figure 2 could be implemented.

Figure 2, the point is well taken. We have revised the figure which is now divided in two figures for better clarity and visibility. The log rank for each gene is provided in the figure legend.

There is a mistake in citation no. 9 and 10. These two overlaps. Please, separate them.

Thank you for the comment, we have separated the citations as per suggestion.

Round 2

Reviewer 2 Report

Authors answered all comments and suggestions.

OK